# Diffusion Models for Graphs Benefit From Discrete State Spaces

**Kilian Haefeli**[1]    **Karolis Martinkus**[1*]  **Nathanaël Perraudin**[2*]  **Roger Wattenhofer**[1]

[1]ETH Zurich    [2]Swiss Data Science Center

{khaefeli,martinkus,wattenhofer}@ethz.ch, nathanael.perraudin@sdsc.ethz.ch

## Abstract

Denoising diffusion probabilistic models and score-matching models have proven to be very powerful for generative tasks. While these approaches have also been applied to the generation of discrete graphs, they have, so far, relied on continuous Gaussian perturbations. Instead, in this work, we suggest using discrete noise for the forward Markov process. This ensures that in every intermediate step the graph remains discrete. Compared to the previous approach, our experimental results on four datasets and multiple architectures show that using a discrete noising process results in higher quality generated samples indicated with an average MMDs reduced by a factor of 1.5. Furthermore, the number of denoising steps is reduced from 1000 to 32 steps leading to 30 times faster sampling procedure.

## 1 Introduction

Score-based [1] and denoising diffusion probabilistic models (DDPMs) [2, 3] have recently achieved striking results in generative modeling and in particular in image generation. Instead of learning a complex model that generates samples in a single pass (like a Generative Adversarial Network [4] (GAN) or a Variational Auto-Encoder [5] (VAE)), a diffusion model is a parameterized Markov Chain trained to reverse an iterative predefined process that gradually transforms a sample into pure noise. Although diffusion processes have been proposed for both continuous [6] and discrete [7] state spaces, their use for graph generation has only focused on Gaussian diffusion processes which operate in the continuous state space [8, 9]. We believe that using a continuous diffusion process to generate a discrete adjacency matrix is sub-optimal as a significant part of the model expressive power will be wasted in learning to generate "discrete-like" outputs. Instead, a discrete noising process forces each intermediary step of the chain to be a "valid" graph.

In this contribution, we follow the Discrete DDPM procedure proposed by Austin et al. [7], Hoogeboom et al. [10] and obtained forward noising process that leads to random Erdős–Rényi graphs [11]. Our experiments show that using discrete noise indeed greatly reduces the number of denoising steps that are needed and improves the sample quality. We also suggest the use of a simple expressive graph neural network architecture [12] for denoising, which, while bringing expressivity benefits, contrasts with more complicated architectures currently used for graph denoising [8].

## 2 Related Work

Traditionally, graph generation has been studied through the lens of random graph models [11, 13, 14]. However, due to the complexity of the graph generation problem, deep generative models have achieved better results. The most successful graph generative models can be divided into two different techniques: a) auto-regressive graph generative models, which generate the graph sequentially node-by-node [15, 16]; b) one-shot generative models which generate the whole graph in a single forward pass [8, 9, 17–21]. While auto-regressive models can generate graphs with hundreds or even thousands of nodes, they can suffer from mode collapse [20, 21]. Finally, graph Variational Auto

---

*Equal contribution.

K. Haefeli et al., Diffusion Models for Graphs Benefit From Discrete State Spaces (Extended Abstract). Presented at the First Learning on Graphs Conference (LoG 2022), Virtual Event, December 9–12, 2022.

Encoders (VAE) remain difficult to train, as the loss function needs to be permutation invariant [22] which can necessitate an expensive graph matching step [17].

In contrast, the score-based models [8, 9] have the potential to provide both, a simple, stable training objective similar to the auto-regressive models and good graph distribution coverage provided by the one-shot models. Niu et al. [8] provided the first score-based model for graph generation (directly using the score-based model formulation by Song and Ermon [1]). Jo et al. [9] extended this to featured graph generation, which lead to promising results for molecule generation. Importantly, both contributions rely on a continuous Gaussian noise process and use a thousand denoising steps to achieve good results, which makes for a slow graph generation.

As shown by Song et al. [6], score matching is tightly related to denoising diffusion probabilistic models [3] which provide a more flexible formulation, more easily amendable for graph generation. In particular, for the noisy samples to remain discrete graphs, the perturbations need to be discrete. Discrete diffusion, using multinomial distribution, was proposed in Hoogeboom et al. [10] and then extended in Austin et al. [7]. It has been successfully used for quantized image generation [23, 24] and text generation [25]. A new, concurrent work by Vignac et al. [26] also investigates discrete DDPM for graph generation and confirms the benefits we outline in this paper.

## 3 Discrete Diffusion for Simple Graphs

Diffusion models [2] are generative models based on a forward and a reverse Markov process. The forward process, denoted $q(\boldsymbol{A}_{1:T} \mid \boldsymbol{A}_0) = \prod_{t=1}^{T} q(\boldsymbol{A}_t \mid \boldsymbol{A}_{t-1})$ generates a sequence of increasingly noisier latent variables $\boldsymbol{A}_t$ from the initial sample $\boldsymbol{A}_0$, to white noise $\boldsymbol{A}_T$. Here the sample $\boldsymbol{A}_0$ and the latent variables $\boldsymbol{A}_t$ are adjacency matrices. The learned reverse process $p_\theta(\boldsymbol{A}_{1:T}) = p(\boldsymbol{A}_T) \prod_{t=1}^{T} q(\boldsymbol{A}_{t-1} \mid \boldsymbol{A}_t)$ attempts to progressively denoise the latent variable $\boldsymbol{A}_t$ in order to produce samples from the desired distribution. Here we will focus on simple graphs, but the approach can be extended in a straightforward manner to account for different edge types. We use the model from [10] and, for convenience, adopt the representation of [7] for our discrete process.

### 3.1 Forward Process

Let the row vector $\boldsymbol{a}_t^{ij} \in \{0, 1\}^2$ be the one-hot encoding of $i, j$ element of the adjacency matrix $\boldsymbol{A}_t$. Here $t \in [0, T]$ denotes the timestep of the process, where $\boldsymbol{A}_0$ is a sample from the data distribution and $\boldsymbol{A}_T$ is an Erdős–Rényi random graph. The forward process is described as repeated multiplication of each adjacency element type row vector $q(\boldsymbol{a}_t^{ij}|\boldsymbol{a}_t^{ij}) = \mathrm{Cat}(\boldsymbol{a}_t^{ij}|p = \boldsymbol{a}_{t-1}^{ij}\boldsymbol{Q}_t$ with a double stochastic matrix $\boldsymbol{Q}_t$. Note that the forward process is independent for each edge/non-edge $i \neq j$. The matrix $\boldsymbol{Q}_t \in \mathbb{R}^{2 \times 2}$ is modeled as

$$\boldsymbol{Q}_t = \begin{bmatrix} 1 - \beta_t & \beta_t \\ \beta_t & 1 - \beta_t \end{bmatrix}, \tag{1}$$

where $\beta_t$ is the probability of not changing the edge state[2]. This formulation has the advantage to allow direct sampling at any timestep of the diffusion process without computing any previous timesteps. Indeed the matrix $\overline{\boldsymbol{Q}}_t = \prod_{i<t} \boldsymbol{Q}_i$ can be expressed in the form of (1) with $\beta_t$ being replaced by $\overline{\beta}_t = \frac{1}{2} - \frac{1}{2} \prod_{i<t}(1 - 2\beta_i)$. Eventually, we want the probability $\overline{\beta}_t \in [0, 0.5]$ to vary from 0 (unperturbed sample) to 0.5 (pure noise). In this contribution, we limit ourselves to symmetric graphs and therefore only need to model the upper triangular part of the adjacency matrix. The noise is sampled i.i.d. over all of the edges.

### 3.2 Reverse Process

To sample from the data distribution, the forward process needs to be reversed. Therefore, we need to estimate $q(\boldsymbol{A}_{t-1}|\boldsymbol{A}_t, \boldsymbol{A}_0)$. In our case, using the Markov property of the forward process this can be rewritten as (see Appendix A for derivation):

$$q(\boldsymbol{A}_{t-1}|\boldsymbol{A}_t, \boldsymbol{A}_0) = q(\boldsymbol{A}_t|\boldsymbol{A}_{t-1}) \frac{q(\boldsymbol{A}_{t-1}|\boldsymbol{A}_0)}{q(\boldsymbol{A}_t|\boldsymbol{A}_0).} \tag{2}$$

Note that (2) is entirely defined by $\beta_t$ and $\overline{\beta}_t$ and $\boldsymbol{A}_0$ (see Appendix A, Equation 4).

---

[2]Note that two different $\beta$'s could be used for edges and non-edges. This case is left for future work.

### 3.3 Loss

Diffusion models are typically trained to minimize a variational upper bound on the negative log-likelihood. This bound can be expressed as (see Appendix C or [3, Equation 5]):

$$
L_{\text{vb}}(\boldsymbol{A}_0)) := \mathbb{E}_{q(\boldsymbol{A}_0)} \Bigg[ \underbrace{D_{KL}(q(\boldsymbol{A}_T|\boldsymbol{A}_0)\|p_\theta(\boldsymbol{A}_T))}_{L_T}
$$

$$
+ \sum_{t=1}^{T} \mathbb{E}_{q(\boldsymbol{A}_t|\boldsymbol{A}_0)} \underbrace{D_{KL}(q(\boldsymbol{A}_{t-1}|\boldsymbol{A}_t,\boldsymbol{A}_0)\|p_\theta(\boldsymbol{A}_{t-1}|\boldsymbol{A}_t))}_{L_t} \underbrace{-\mathbb{E}_{q(\boldsymbol{A}_1|\boldsymbol{A}_0)} \log(p_\theta(\boldsymbol{A}_0|\boldsymbol{A}_1))}_{L_0} \Bigg]
$$

Practically, the model is trained to directly minimize the losses $L_t$, i.e. the KL divergence $D_{KL}(q(\boldsymbol{A}_{t-1} \mid \boldsymbol{A}_t,\boldsymbol{A}_0)\|p_\theta(\boldsymbol{A}_{t-1} \mid \boldsymbol{A}_t))$ by using the tractable parametrization of $q(\boldsymbol{A}_{t-1}|\boldsymbol{A}_t,\boldsymbol{A}_0)$ from (2). Note that the discrete setting of the selected noise distribution prevents training the model to approximate the gradient of the distribution as done by score-matching graph generative models [8, 9].

**Parametrization of the reverse process.** While it is possible to predict the logits of $p_\theta(\boldsymbol{A}_{t-1} \mid \boldsymbol{A}_t)$ in order to minimize $L_{\text{vb}}$, we follow [3, 7, 10] and use a network $\text{nn}_\theta(\boldsymbol{A}_t)$ that predict the logits of the distribution $p_\theta(\boldsymbol{A}_0 \mid \boldsymbol{A}_t)$. This parametrization is known to stabilize the training procedure. To minimize $L_{\text{vb}}$, (2) can be used to recover $p_\theta(\boldsymbol{A}_{t-1} \mid \boldsymbol{A}_t)$ from $\boldsymbol{A}_0$ and $\boldsymbol{A}_t$.

**Alternate loss.** Many implementations of DDPMs found it beneficial to use alternative losses. For instance, [3] derived a simplified loss function that reweights the ELBO. Hybrid losses have been used in [27] and [7]. As shown in Appendix D, it is possible to use the parametrization $p_\theta(\boldsymbol{A}_0 \mid \boldsymbol{A}_t)$, i.e. to replace the KL term $L_t$ with $L_t = -\log(p_\theta(\boldsymbol{A}_0 \mid \boldsymbol{A}_t))$. Empirically, we found that minimizing

$$
L_{\text{simple}} := -\mathbb{E}_{q(\boldsymbol{A}_0)} \sum_{t=1}^{T} \left( 1 - 2 \cdot \overline{\beta}_t + \frac{1}{T} \right) \cdot \mathbb{E}_{q(\boldsymbol{A}_t|\boldsymbol{A}_0)} \log p_\theta(\boldsymbol{A}_0 \mid \boldsymbol{A}_t)) \tag{3}
$$

leads to stable training and better results. Note that this loss equals the cross-entropy loss between $\boldsymbol{A}_0$ and $\text{nn}_\theta(\boldsymbol{A}_t)$. The re-weighting $1 - 2 \cdot \overline{\beta}_t + \frac{1}{T}$, which assigns linearly more importance to the less noisy samples, has been proposed in [23, Equation 7].

### 3.4 Sampling

For each loss, we used a specific sampling algorithm. For both approaches, we start by sampling each edge independently from a Bernoulli distribution with probability $p = 1/2$ (Erdős–Rényi random graph). Then, for the $L_{\text{vb}}$ loss we follow Ho et al. [3] and iteratively reverse the chain by sampling Bernoulli-sampling from $p_\theta(\boldsymbol{A}_{t-1} \mid \boldsymbol{A}_t)$ until we obtain at our sample of $p_\theta(\boldsymbol{A}_0 \mid \boldsymbol{A}_1)$. For the loss function $L_{\text{simple}}$, we sample $\boldsymbol{A}_0$ directly from $p_\theta(\boldsymbol{A}_0|\boldsymbol{A}_t)$ for each step t and obtain $\boldsymbol{A}_{t-1}$ by sampling again from $q(\boldsymbol{A}_{t-1} \mid \boldsymbol{A}_0)$. The two approaches are described algorithmically in Appendix E.

The values of $\overline{\beta}_t$ are selected following a simple linear schedule for our reverse process [2] . We found it works similarly well as other options such as cosine schedule [27]. Note that in this case $\beta_t$ can be obtained from $\overline{\beta}_t$ in a straightforward manner (see Appendix B).

## 4 Experiments

We compare our graph discrete diffusion approach to the original score-based approach proposed by Niu et al. [8]. Models using this original formulation are denoted by *score*. We follow the training and evaluation setup used by previous contributions [8, 9, 15, 19]. More details can be found in Appendix G. For evaluation, we compute MMD metrics from [15] between the generated graphs and the test set, namely, the degree distribution, the clustering coefficient, and the 4-node orbit counts. To demonstrate the efficiency of the discrete parameterization, the discrete models only use 32 denoising steps, while the score-based models use 1000 denoising steps, as originally proposed. We compare two architectures: 1. EDP-GNN as introduced by Niu et al. [8], and 2. a simpler and more expressive

provably powerful graph network (PPGN) [12]. See Appendix F for a more detailed description of the architectures.

Table 1 shows the results for two datasets, Community-small ($12 \leq n \leq 20$) and Ego-small ($4 \leq n \leq 18$), used by Niu et al. [8]. To better compare our approach to traditional score-based graph generation, in Table 2, we additionally perform experiments on slightly more challenging datasets with larger graphs. Namely, a stochastic-block-model (SBM) dataset with three communities, which in total consists of ($24 \leq n \leq 27$) nodes and a planar dataset with ($n = 60$) nodes. Detailed information on the datasets can be found in Appendix H. Additional details concerning the evaluation setup are provided in Appendix G.4.

|  | Community | | | | Ego | | | | |
| Model | Deg. | Clus. | Orb. | Avg. | Deg. | Clus. | Orb. | Avg. | Total |
| --- | --- | --- | --- | --- | --- | --- | --- | --- | --- |
| GraphRNN† | 0.030 | 0.030 | 0.010 | **0.017** | 0.040 | 0.050 | 0.060 | 0.050 | 0.033 |
| GNF† | 0.120 | 0.150 | 0.020 | 0.097 | 0.010 | 0.030 | 0.001 | 0.014 | 0.055 |
| EDP-Score† | 0.006 | 0.127 | 0.018 | 0.050 | 0.010 | 0.025 | 0.003 | **0.013** | 0.031 |
| SDE-Score† | 0.045 | 0.086 | 0.007 | 0.046 | 0.021 | 0.024 | 0.007 | 0.017 | 0.032 |
| EDP-Score[3] | 0.016 | 0.810 | 0.110 | 0.320 | 0.04 | 0.064 | 0.005 | 0.037 | 0.178 |
| PPGN-Score | 0.081 | 0.237 | 0.284 | 0.200 | 0.019 | 0.049 | 0.005 | 0.025 | 0.113 |
| PPGN $L_{\text{vb}}$ | 0.023 | 0.061 | 0.015 | 0.033 | 0.025 | 0.039 | 0.019 | 0.027 | 0.03 |
| PPGN $L_{\text{simple}}$ | 0.019 | 0.044 | 0.005 | 0.023 | 0.018 | 0.026 | 0.003 | 0.016 | **0.019** |
| EDP $L_{\text{simple}}$ | 0.024 | 0.04 | 0.012 | 0.026 | 0.019 | 0.031 | 0.017 | 0.022 | 0.024 |

**Table 1:** MMD results for the Community and the Ego datasets. All values are averaged over 5 runs with 1024 generated samples without any sub-selection. The "Total" column denotes the average MMD over all of the 6 measurements. The best results of the "Avg." and "Total" columns are shown in bold. † marks the results taken from the original papers.

**Results.** In Table 1, we observe that the proposed discrete diffusion process using the $L_{\text{vb}}$ loss and PPGN model leads to slightly improved average MMDs over the competitors. The $L_{\text{simple}}$ loss further improve the result over $L_{\text{vb}}$. The fact that the EDP-$L_{\text{simple}}$ model has significantly lower

|  | SBM-27 | | | | Planar-60 | | | | |
| Model | Deg. | Clus. | Orb. | Avg. | Deg. | Clus. | Orb. | Avg. | Total |
| --- | --- | --- | --- | --- | --- | --- | --- | --- | --- |
| EDP-Score | 0.014 | 0.800 | 0.190 | 0.334 | 1.360 | 1.904 | 0.534 | 1.266 | 0.8 |
| PPGN $L_{\text{simple}}$ | 0.007 | 0.035 | 0.072 | **0.038** | 0.029 | 0.039 | 0.036 | **0.035** | **0.036** |
| EDP $L_{\text{simple}}$ | 0.046 | 0.184 | 0.064 | 0.098 | 0.017 | 1.928 | 0.785 | 0.910 | 0.504 |

**Table 2:** MMD results for the SBM-27 and the Planar-60 datasets.

MMD values than the EDP-score model is a strong indication that the proposed loss and the discrete formulation are the cause of the improvement rather than the PPGN architecture. This improvement comes with the additional benefit that sampling is greatly accelerated (30 times) as the number of timesteps is reduced from 1000 to 32. Table 2 shows that the proposed discrete formulation is even more beneficial when graph size and complexity increase. The PPGN-Score even becomes infeasible to run in this setting, due to the prohibitively expensive sampling procedure. A qualitative evaluation of the generated graphs is performed in Appendix I. Visually, the $L_{\text{simple}}$ loss leads to the best samples.

To further showcase the performance improvement of using discrete diffusion we performed a study on how the number of sampling steps influences generated sample quality for PPGN $L_{\text{simple}}$, which uses discrete noise and PPGN-Score, which uses Gaussian noise. In Figure 1 we can see that our model using discrete noise already achieves the best generation quality with just 48 denoising steps, while the model with Gaussian noise achieves worse results even after 1024 steps.

## 5 Conclusion

In this work, we demonstrated that discrete diffusion can increase sample quality and greatly improve the efficiency of denoising diffusion for graph generation. While the approach was presented for simple graphs with non-attributed edges, it could also be extended to cover graphs with edge attributes.

**Figure 1:** Average MMD compared to the number of denoising steps used on the Ego dataset for PPGN $L_{\text{simple}}$, which uses discrete noise and PPGN-Score, which uses Gaussian noise.

---

[3]The discrepancy with the EDP-Score† results comes from the fact that using the code provided by the authors, we were unable to reproduce their results. Strangely, their code leads to good results when used with our discrete formulation and $L_{\text{simple}}$ loss improving over the result reported in their contribution.

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

## A    Reverse Process Derivations

In this appendix, we provide the derivation of the reverse probability $q(\boldsymbol{A}_{t-1}|\boldsymbol{A}_t, \boldsymbol{A}_0)$. Using the Bayes rule, we obtain

$$
\begin{aligned}
q(\boldsymbol{A}_{t-1}|\boldsymbol{A}_t, \boldsymbol{A}_0) &= \frac{q(\boldsymbol{A}_t \mid \boldsymbol{A}_{t-1}, \boldsymbol{A}_0) \cdot q(\boldsymbol{A}_{t-1}, \boldsymbol{A}_0)}{q(\boldsymbol{A}_t, \boldsymbol{A}_0)} \\
&= \frac{q(\boldsymbol{A}_t \mid \boldsymbol{A}_{t-1}) \cdot q(\boldsymbol{A}_{t-1} \mid \boldsymbol{A}_0) q(\boldsymbol{A}_0)}{q(\boldsymbol{A}_t \mid \boldsymbol{A}_0) \cdot q(\boldsymbol{A}_0)} \\
&= q(\boldsymbol{A}_t \mid \boldsymbol{A}_{t-1}) \cdot \frac{q(\boldsymbol{A}_{t-1} \mid \boldsymbol{A}_0)}{q(\boldsymbol{A}_t \mid \boldsymbol{A}_0)},
\end{aligned}
$$

where we use the fact that $q(\boldsymbol{A}_t \mid \boldsymbol{A}_{t-1}, \boldsymbol{A}_0) = q(\boldsymbol{A}_t \mid \boldsymbol{A}_{t-1})$ since $\boldsymbol{A}_t$ is independent of $\boldsymbol{A}_0$ given $\boldsymbol{A}_{t-1}$.

This reverse probability is entirely defined with $\beta_t$ and $\bar{\beta}_t$. For the $i, j$ element of $\boldsymbol{A}$ (denoted $\boldsymbol{A}^{ij}$), we obtain:

$$
q(\boldsymbol{A}_{t-1}^{ij} = 1|\boldsymbol{A}_t^{ij}, \boldsymbol{A}_0^{ij}) = \begin{cases} (1 - \beta_t) \cdot \frac{(1-\bar{\beta}_{t-1})}{1-\bar{\beta}_t}, & \text{if} \boldsymbol{A}_t^{ij} = 1, \boldsymbol{A}_0^{ij} = 1 \\ (1 - \beta_t) \cdot \frac{\bar{\beta}_{t-1}}{\bar{\beta}_t}, & \text{if } \boldsymbol{A}_t^{ij} = 1, \boldsymbol{A}_0^{ij} = 0 \\ \beta_t \cdot \frac{(1-\bar{\beta}_{t-1})}{\bar{\beta}_t}, & \text{if } \boldsymbol{A}_t^{ij} = 0, \boldsymbol{A}_0^{ij} = 1 \\ \beta_t \cdot \frac{\bar{\beta}_{t-1}}{1-\bar{\beta}_t}, & \text{if } \boldsymbol{A}_t^{ij} = 0, \boldsymbol{A}_0^{ij} = 0 \end{cases} \tag{4}
$$

## B    Conversion of $\overline{\beta}_t$ to $\beta_t$

The selected linear schedule provides us with the values of $\overline{\beta}_t$. In this appendix, we compute an expression for $\beta_t$ from $\overline{\beta}_t$, which allows us easy computation of (2). By definition, we have $\overline{\boldsymbol{Q}}_t = \overline{\boldsymbol{Q}}_{t-1} \boldsymbol{Q}_t$ which is equivalent to

$$
\begin{pmatrix} 1 - \bar{\beta}_{t-1} & \bar{\beta}_{t-1} \\ \bar{\beta}_{t-1} & 1 - \bar{\beta}_{t-1} \end{pmatrix} \begin{pmatrix} 1 - \beta_t & \beta_t \\ \beta_t & 1 - \beta_t \end{pmatrix} = \begin{pmatrix} 1 - \bar{\beta}_t & \bar{\beta}_t \\ \bar{\beta}_t & 1 - \bar{\beta}_t \end{pmatrix}
$$

Let us select the first row and first column equality. We obtain the following equation

$$
\left(1 - \bar{\beta}_{t-1}\right)\left(1 - \beta_t\right) + \bar{\beta}_{t-1} \beta_t = 1 - \bar{\beta}_t,
$$

which, after some arithmetic, provides us with the desired answer

$$
\beta_t = \frac{\bar{\beta}_{t-1} - \bar{\beta}_t}{2\bar{\beta}_{t-1} - 1}.
$$

## C    ELBO derivation

The general Evidence Lower Bound (ELBO) formula states that

$$
\log\left(p_\theta\left(x\right)\right) \geq \mathbb{E}_{z \sim q}\left[\log\left(\frac{p\left(x, z\right)}{q\left(z\right)}\right)\right]
$$

for any distribution $q$ and latent $z$. In our case, we use $\boldsymbol{A}_{1:T}$ as a latent variable and obtain

$$
-\log\left(p_\theta\left(\boldsymbol{A}_0\right)\right) \leq \mathbb{E}_{\boldsymbol{A}_{1:T} \sim q(\boldsymbol{A}_{1:T}|\boldsymbol{A}_0)}\left[\log\left(\frac{p_\theta\left(\boldsymbol{A}_{0:T}\right)}{q\left(\boldsymbol{A}_{1:T} \mid \boldsymbol{A}_0\right)}\right)\right] := L_{\text{vb}}(\boldsymbol{A}_0)
$$

We use $L_{\text{vb}} = \mathbb{E}\left[L_{\text{vb}}(\boldsymbol{A}_0)\right)]$ and obtain

$$
L_{\text{vb}} = \mathbb{E}_{q(\boldsymbol{A}_{0:T})}\left[-\log\left(\frac{p_\theta(\boldsymbol{A}_{0:T})}{q(\boldsymbol{A}_{1:T}\mid\boldsymbol{A}_0)}\right)\right]
$$

$$
= \mathbb{E}_q\left[-\log(p_\theta(\boldsymbol{A}_T)) - \sum_{t=1}^{T}\log\left(\frac{p_\theta(\boldsymbol{A}_{t-1}\mid\boldsymbol{A}_t)}{q(\boldsymbol{A}_t\mid\boldsymbol{A}_{t-1})}\right)\right]
$$

$$
= \mathbb{E}_q\left[-\log(p_\theta(\boldsymbol{A}_T)) - \sum_{t=2}^{T}\log\left(\frac{p_\theta(\boldsymbol{A}_{t-1}\mid\boldsymbol{A}_t)}{q(\boldsymbol{A}_t\mid\boldsymbol{A}_{t-1})}\right) - \log\left(\frac{p_\theta(\boldsymbol{A}_0\mid\boldsymbol{A}_1)}{q(\boldsymbol{A}_1\mid\boldsymbol{A}_0)}\right)\right]
$$

$$
= \mathbb{E}_q\left[-\log(p_\theta(\boldsymbol{A}_T)) - \sum_{t=2}^{T}\log\left(\frac{p_\theta(\boldsymbol{A}_{t-1}\mid\boldsymbol{A}_t)}{q(\boldsymbol{A}_{t-1}\mid\boldsymbol{A}_t,\boldsymbol{A}_0)}\cdot\frac{q(\boldsymbol{A}_{t-1}\mid\boldsymbol{A}_0)}{q(\boldsymbol{A}_t\mid\boldsymbol{A}_0)}\right) - \log\left(\frac{p_\theta(\boldsymbol{A}_0\mid\boldsymbol{A}_1)}{q(\boldsymbol{A}_1\mid\boldsymbol{A}_0)}\right)\right]
$$

$$
(5)
$$

$$
= \mathbb{E}_q\left[-\log\left(\frac{p_\theta(\boldsymbol{A}_T)}{q(\boldsymbol{A}_T\mid\boldsymbol{A}_0)}\right) - \sum_{t=2}^{T}\log\left(\frac{p_\theta(\boldsymbol{A}_{t-1}\mid\boldsymbol{A}_t)}{q(\boldsymbol{A}_{t-1}\mid\boldsymbol{A}_t,\boldsymbol{A}_0)}\right) - \log(p_\theta(\boldsymbol{A}_0\mid\boldsymbol{A}_1))\right]
$$

$$
= \mathbb{E}_{\mathbb{E}_{q(\boldsymbol{A}_0)}}\left[D_{KL}(q(\boldsymbol{A}_T\mid\boldsymbol{A}_0)\|p_\theta(\boldsymbol{A}_T)) + \sum_{t=2}^{T}\mathbb{E}_{q(\boldsymbol{A}_t\mid\boldsymbol{A}_0)}D_{KL}(q(\boldsymbol{A}_{t-1}\mid\boldsymbol{A}_t,\boldsymbol{A}_0)\|p_\theta(\boldsymbol{A}_{t-1}\mid\boldsymbol{A}_t))\right.
$$

$$
\left.-\mathbb{E}_{q(\boldsymbol{A}_1\mid\boldsymbol{A}_0)}\log(p_\theta(\boldsymbol{A}_0\mid\boldsymbol{A}_1))\right]
$$

where (5) follows from

$$
q(\boldsymbol{A}_{t-1}\mid\boldsymbol{A}_t,\boldsymbol{A}_0) = \frac{q(\boldsymbol{A}_t\mid\boldsymbol{A}_{t-1},\boldsymbol{A}_0)\,q(\boldsymbol{A}_{t-1},\boldsymbol{A}_0)}{q(\boldsymbol{A}_t,\boldsymbol{A}_0)}
$$

$$
= \frac{q(\boldsymbol{A}_t\mid\boldsymbol{A}_{t-1})\,q(\boldsymbol{A}_{t-1}\mid\boldsymbol{A}_0)}{q(\boldsymbol{A}_t\mid\boldsymbol{A}_0)}.
$$

The ELBO loss can be optimized by optimizing each of the $D_{KL}(q(\boldsymbol{A}_{t-1}\mid\boldsymbol{A}_t,\boldsymbol{A}_0)\|p_\theta(\boldsymbol{A}_{t-1}\mid\boldsymbol{A}_t))$ terms corresponding to different time steps $t$. Since we are dealing with the categorical distributions optimization of $D_{KL}(q(\boldsymbol{A}_{t-1}\mid\boldsymbol{A}_t,\boldsymbol{A}_0)\|p_\theta(\boldsymbol{A}_{t-1}\mid\boldsymbol{A}_t))$ is equivalent to optimizing the cross entropy loss between $q(\boldsymbol{A}_{t-1}\mid\boldsymbol{A}_t,\boldsymbol{A}_0)$ and $p_\theta(\boldsymbol{A}_{t-1}\mid\boldsymbol{A}_t)$. So for training the model, we can select a random time step $t$ and optimize the corresponding KL divergence term using corss entropy loss.

## D   Simple Loss

The simple loss is obtained by taking a slightly different bound to the negative log-likekelyhood (ELBO).

$$
-\log(p_\theta(\boldsymbol{A}_0)) \leq \mathbb{E}_{\boldsymbol{A}_{1:T}\sim q(\boldsymbol{A}_{1:T}\mid\boldsymbol{A}_0)}\left[\log\left(\frac{p_\theta(\boldsymbol{A}_{0:T})}{q(\boldsymbol{A}_{1:T}\mid\boldsymbol{A}_0)}\right)\right]
$$

$$
= \mathbb{E}_q\left[-\log\left(\frac{p_\theta(\boldsymbol{A}_{0:T})}{q(\boldsymbol{A}_{1:T}\mid\boldsymbol{A}_0)}\right)\right]
$$

$$
= \mathbb{E}_q\left[-\log(p_\theta(\boldsymbol{A}_T)) - \sum_{t=2}^{T}\log\left(\frac{p_\theta(\boldsymbol{A}_{t-1}\mid\boldsymbol{A}_t)}{q(\boldsymbol{A}_t\mid\boldsymbol{A}_{t-1})}\right) - \log\left(\frac{p_\theta(\boldsymbol{A}_0\mid\boldsymbol{A}_1)}{q(\boldsymbol{A}_1\mid\boldsymbol{A}_0)}\right)\right]
$$

$$
= \mathbb{E}_q\left[-\log(p_\theta(\boldsymbol{A}_T)) - \log\left(\frac{p_\theta(\boldsymbol{A}_0\mid\boldsymbol{A}_1)}{q(\boldsymbol{A}_1\mid\boldsymbol{A}_0)}\right)\right.
$$

$$
\left.- \sum_{t=2}^{T}\log\left(\frac{q(\boldsymbol{A}_{t-1}\mid\boldsymbol{A}_t,\boldsymbol{A}_0)\,p_\theta(\boldsymbol{A}_0\mid\boldsymbol{A}_t)}{q(\boldsymbol{A}_{t-1}\mid\boldsymbol{A}_t,\boldsymbol{A}_0)}\cdot\frac{q(\boldsymbol{A}_{t-1}\mid\boldsymbol{A}_0)}{q(\boldsymbol{A}_t\mid\boldsymbol{A}_0)}\right)\right]
$$

$$
= \mathbb{E}_q\left[-\log\left(\frac{p_\theta(\boldsymbol{A}_T)}{q(\boldsymbol{A}_T\mid\boldsymbol{A}_0)}\right) - \sum_{t=2}^{T}\log(p_\theta(\boldsymbol{A}_0\mid\boldsymbol{A}_t)) - \log(p_\theta(\boldsymbol{A}_0\mid\boldsymbol{A}_1))\right]
$$

Therefore the simple loss, the term $L_t = D_{KL}(q(\boldsymbol{A}_{t-1}\mid\boldsymbol{A}_t,\boldsymbol{A}_0)\|p_\theta(\boldsymbol{A}_{t-1}\mid\boldsymbol{A}_t))$ is replaced by

$$
L_t = \mathbb{E}_{q(\boldsymbol{A}_t\mid\boldsymbol{A}_0)}\left[\log(p_\theta(\boldsymbol{A}_0\mid\boldsymbol{A}_t))\right]
$$

---

**Algorithm 1** Sampling for $L_{\text{vb}}$

---

1: $\forall i, j | i > j$: $\boldsymbol{A}_T^{ij} \sim \mathcal{B}_{p=1/2}$
2: **for** $t = T, ..., 1$ **do**
3:     Compute $p_\theta(\boldsymbol{A}_{t-1} | \boldsymbol{A}_t)$
4:     $\boldsymbol{A}_{t-1} \sim p_\theta(\boldsymbol{A}_{t-1} | \boldsymbol{A}_t)$
5: **end for**

---

**Algorithm 2** Sampling for $L_{\text{simple}}$

---

1: $\forall i, j | i > j$: $\boldsymbol{A}_T^{ij} \sim \mathcal{B}_{p=1/2}$
2: **for** $t = T, ..., 1$ **do**
3:     $\tilde{\boldsymbol{A}}_0 \sim p_\theta(\boldsymbol{A}_0 | \boldsymbol{A}_t)$
4:     $\boldsymbol{A}_{t-1} \sim q(\boldsymbol{A}_{t-1} | \tilde{\boldsymbol{A}}_0)$
5: **end for**

---

**Sampling algorithms.** To sample a new graph, we start by generating a random Erdős–Rényi graph $\boldsymbol{A}_T$, i.e., each edge is randomly drawn independently with a probability $p = 1/2$. Then, we reverse each step of the Markov chain until we get to $\boldsymbol{A}_0$. Algorithms 1 and 2 differ in how this is done.
In Algorithm 1, we obtain $\boldsymbol{A}_{t-1}$ from $\boldsymbol{A}_t$ by 1. computing edge probabilities using the model $p_\theta(\boldsymbol{A}_{t-1} | \boldsymbol{A}_t)$, and 2., sampling the new adjacency matrix $\boldsymbol{A}_{t-1}$.
In Algorithm 2, we obtain $\boldsymbol{A}_{t-1}$ from $\boldsymbol{A}_t$ by 1. computing edge probabilities of the target adjacency matrix $\boldsymbol{A}_0$ using the model $p_\theta(\boldsymbol{A}_0 | \boldsymbol{A}_t)$, 2. sampling to get an estimate to obtain $\tilde{\boldsymbol{A}}_0$, and 3., sampling the new adjacency matrix $\boldsymbol{A}_{t-1}$ from $q(\boldsymbol{A}_{t-1} | \tilde{\boldsymbol{A}}_0)$.

## E    Sampling Algorithms

Here in Algorithms 1 and 2 we provide an algorithmic description of the two sampling approaches described in Section 3.4. Here $\mathcal{B}_{p=1/2}$ denotes the Bernoulli distribution with parameter $p = 1/2$, which corresponds to the Erdős–Rényi random graph model.

## F    Models

### F.1    Edgewise Dense Prediction Graph Neural Network (EDP-GNN)

The EDP-GNN model introduced by Niu et al. [8] extends GIN [28] to work with multi-channel adjacency matrices. This means that a GIN graph neural network is run on multiple different adjacency matrices (channels) and the different outputs are concatenated to produce new node embeddings:

$$\boldsymbol{X}_c^{(k+1)'} = \widetilde{\boldsymbol{A}}_c^{(k)} \boldsymbol{X}^{(k)} + (1 + \epsilon) \boldsymbol{X}^{(k)},$$

$$\boldsymbol{X}^{(k+1)} = \text{Concat}(\boldsymbol{X}_c^{(k+1)'} \text{ for } c \in \{1, \ldots, C^{(k+1)}\}),$$

where $\boldsymbol{X} \in \mathbb{R}^{n \times h}$ is the node embedding matrix with hidden dimension $h$ and $C^{(k)}$ is the number of channels in the input multi-channel adjacency matrix $\widetilde{\boldsymbol{A}}^{(k)} \in \mathbb{R}^{C^{(k)} \times n \times n}$, at layer $k$. The adjacency matrices for the next layer are produced using the node embeddings:

$$\widetilde{\boldsymbol{A}}_{\cdot, i, j}^{(k+1)} = \text{MLP}(\widetilde{\boldsymbol{A}}_{\cdot, i, j}^{(k)}, \boldsymbol{X}_i, \boldsymbol{X}_j).$$

For the first layer, EDP-GNN computes two adjacency matrix $\widetilde{\boldsymbol{A}}^{(0)}$ channels, original input adjacency $\boldsymbol{A}$ and its inversion $\boldsymbol{1}\boldsymbol{1}^T - \boldsymbol{A}$. For node features, node degrees are used $\boldsymbol{X}^{(0)} = \sum_i \boldsymbol{A}_i$.

To produce the final outputs, the outputs of all intermediary layers are concatenated:

$$\widetilde{\boldsymbol{A}} = \text{MLP}_{\text{out}}(\text{Concat}(\widetilde{\boldsymbol{A}}^{(k)} \text{ for } k \in \{1, \ldots, K\})).$$

The final layer always has only one output channel, such that $\boldsymbol{A}_{(t)} = \text{EDP-GNN}(\boldsymbol{A}_{(t-1)})$.

To condition the model on the given noise level $\overline{\beta}_t$, noise-level-dependent scale and bias parameters $\boldsymbol{\alpha}_t$ and $\boldsymbol{\gamma}_t$ are introduced to each layer $f$ of every MLP:

$$f(\widetilde{\boldsymbol{A}}_{\cdot, i, j}) = \text{activation}((\boldsymbol{W} \widetilde{\boldsymbol{A}}_{\cdot, i, j} + \boldsymbol{b})\boldsymbol{\alpha}_t + \boldsymbol{\gamma}_t).$$

### F.2    Provably Powerful Graph Network (PPGN)

The input to the PPGN model used is the adjacency matrix $\boldsymbol{A}_t$ concatenated with the diagonal matrix $\overline{\beta}_t \cdot \boldsymbol{I}$, resulting in an input tensor $\boldsymbol{X}_{in} \in \mathbb{R}^{n \times n \times 2}$. The output tensor is $\boldsymbol{X}_{out} \in \mathbb{R}^{n \times n \times 1}$, where

each $[\boldsymbol{X}_{out}]_{ij}$ represents $p([\boldsymbol{A}_0]_{ij} \mid [\boldsymbol{A}_t]_{ij})$.

Our PPGN implementation, which closely follows Maron et al. [12] is structured as follows:
Let $\boldsymbol{P}$ denote the PPGN model, then

$$\boldsymbol{P}(\boldsymbol{X}_{in}) := (l_{\text{out}} \circ C)(\boldsymbol{X}_{in}) \tag{6}$$

$$C : \mathbb{R}^{n \times n \times 2} \to \mathbb{R}^{n \times n \times (d \cdot h)} \tag{7}$$

$$C(\boldsymbol{X}_{in}) := \text{Concat}((B_d \circ ... \circ B_1)(\boldsymbol{X}_{in}), (B_{d-1} \circ ... \circ B_1)(\boldsymbol{X}_{in}), ..., B_1(\boldsymbol{X}_{in})) \tag{8}$$

The set $\{B_1, ..., B_d\}$ is a set of d different powerful layers implemented as proposed by Maron et al. [12]. We let the input run through different amounts of these powerful layers and concatenate their respective outputs to one tensor of size $n \times n \times (d \cdot h)$. These powerful layers are functions of size:

$$\forall B_i \in \{B_2, ..., B_d\}, B_i : \mathbb{R}^{n \times n \times h} \to \mathbb{R}^{n \times n \times h} \tag{9}$$

$$B_1 : \mathbb{R}^{n \times n \times 1} \to \mathbb{R}^{n \times n \times h}. \tag{10}$$

Finally, we use an MLP 2 to reduce the dimensionality of each matrix element down to 1, so that we can treat the output as an adjacency matrix.

$$l_{\text{out}} : \mathbb{R}^{d \cdot h} \to \mathbb{R}^1, \tag{11}$$

where $l_{\text{out}}$ is applied to each element $[C(\boldsymbol{X}_{in})]_{i,j,.}$ of the tensor $C(\boldsymbol{X}_{in})$ over all its $d \cdot h$ channels. It is used to reduce the number of channels down to a single one which represents $p(\boldsymbol{A}_0|\boldsymbol{A}_t)$.

## G   Training Setup

### G.1   EDP-GNN

The model training setup and hyperparameters used for the EDP-GNN were directly taken from [8]. We used 4 message-passing steps for each GIN, then stacked 5 EDP-GNN layers, for which the maximum number of channels is always set to 4 and the maximum number of node features is 16. We use 32 denoising steps for all datasets besides Planar-60, where we used 256. Opposed to 6 noise levels with 1000 sample steps per level as in the Score-based approach.

### G.2   PPGN

The PPGN model we used for the Ego-small, Community-small, and SBM-27 datasets consists of 6 layers $\{B_1, ..., B_6\}$. After each powerful layer, we apply an instance normalization. The hidden dimension was set to 16. For the Planar-60 dataset, we have used 8 layers and a hidden dimension of 128. We used a batch size of 64 for all datasets and used the Adam optimizer with parameters chosen as follows: learning rate is $0.001$, betas are $(0.9, 0.999)$ and weight decay is $0.999$.

### G.3   Model Selection

We performed a simple model selection where the model which achieves the best training loss is saved and used to generate graphs for testing. We also investigated the use of a validation split and computation of MMD scores versus this validation split for model selection, but we did not find this to produce better results while adding considerable computational overhead.

### G.4   Additional Details on Experimental Setup

Here we provide some details concerning the experimental setup for the results in Tables 1, 2 and Figure 1.

**Details for MMD results in Table 1:**   From the original paper Niu et al. [8], we are unsure if the GNF, GraphRNN, and EDP-Score model selection were used or not. The SDE-Score results in the first section are sampled after training for 5000 epochs and no model selection was used. Due to the compute limitations on the PPGN model, the results for PPGN $L_{\text{vb}}$ are taken after epoch 900 instead of 5000, as results for SDE-Score and EDP-Score have been. The results for PPGN $L_{\text{simple}}$ and EDP $L_{\text{simple}}$ were trained for 2500 epochs.

**Details for MMD results in Table 2:** All results using the EDP-GNN model are trained until epoch 5000 and the PPGN implementation was trained until epoch 2500.

**Details for ablation results in Figure 1:** Experiments were performed on ego-small using 4 different seeds and training one model per seed. Each model was trained for 2500 epochs and no model selection was used. Both implementations used the PPGN model, one based on the score framework and one on our discrete diffusion. For every model, we sampled 256 graphs for which the average of the three MMD metrics (Degree, Clustering, Orbital) is reported. The plot shows the mean and standard deviation of this average MMD over the four seeds.

## H  Datasets

In this appendix, we describe the 4 datasets used in our experiments.

**Ego-small:** This dataset is composed of 200 graphs of 4-18 nodes from the Citeseer network (Sen et al. [29]). The dataset is available in the repository[4] of Niu et al. [8].

**Community-small:** This dataset consists of 100 graphs from 12 to 20 nodes. The graphs are generated in two steps. First, two communities of equal size are generated using the Erdos-Rényi model [11] with parameter $p = 0.7$. Then edges are randomly added between the nodes of the two communities with a probability $p = 0.05$. The dataset is directly taken from the repository of Niu et al. [8].

**SBM-27:** This dataset consists of 200 graphs with 24 to 27 nodes generated using the Stochastic-Block-Model (SBM) with three communities. We use the implementation provided by Martinkus et al. [21]. The parameters used are $p_{intra} = 0.85$, $p_{inter}$=0.046875, where $p_{intra}$ stands for the intra-community (i.e. for a node within the same community) edge probability and $p_{inter}$ stands for the inter-community (i.e. for nodes from different communities) edge probability. The number of nodes for the 3 communities is randomly drawn from $\{7, 8, 9\}$. In expectation, these parameters generate 3 edges between each pair of communities.

**Planar-60:** This dataset consists of 200 randomly generated planar graphs of 60 nodes. We use the implementation provided by Martinkus et al. [21]. To generate a graph, 60 points are first randomly uniformly sampled on the $[0, 1]^2$ plane. Then the graph is generated by applying Delaunay triangulation to these points [30].

## I  Visualization of Sampled Graphs

In the following pages, we provide a visual comparison of graphs generated by the different models.

---

[4]https://github.com/ermongroup/GraphScoreMatching

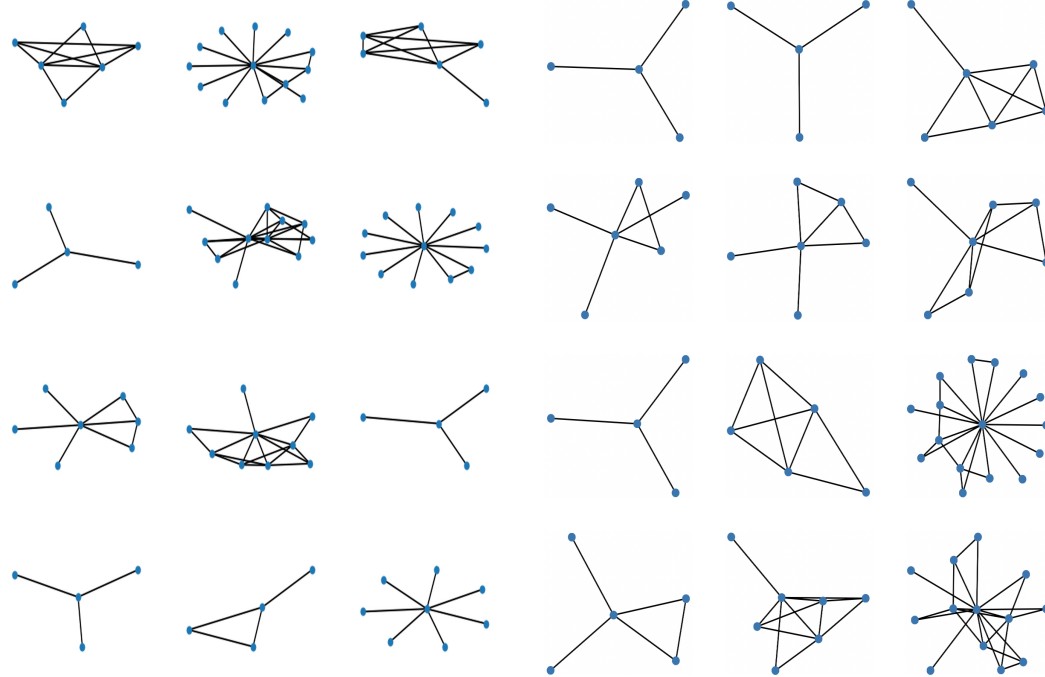

**Figure 2:** Sample graphs from the training set of Ego-small dataset.

**Figure 3:** Sample graphs generated with the model EDP-Score [8] for the Ego-small dataset.

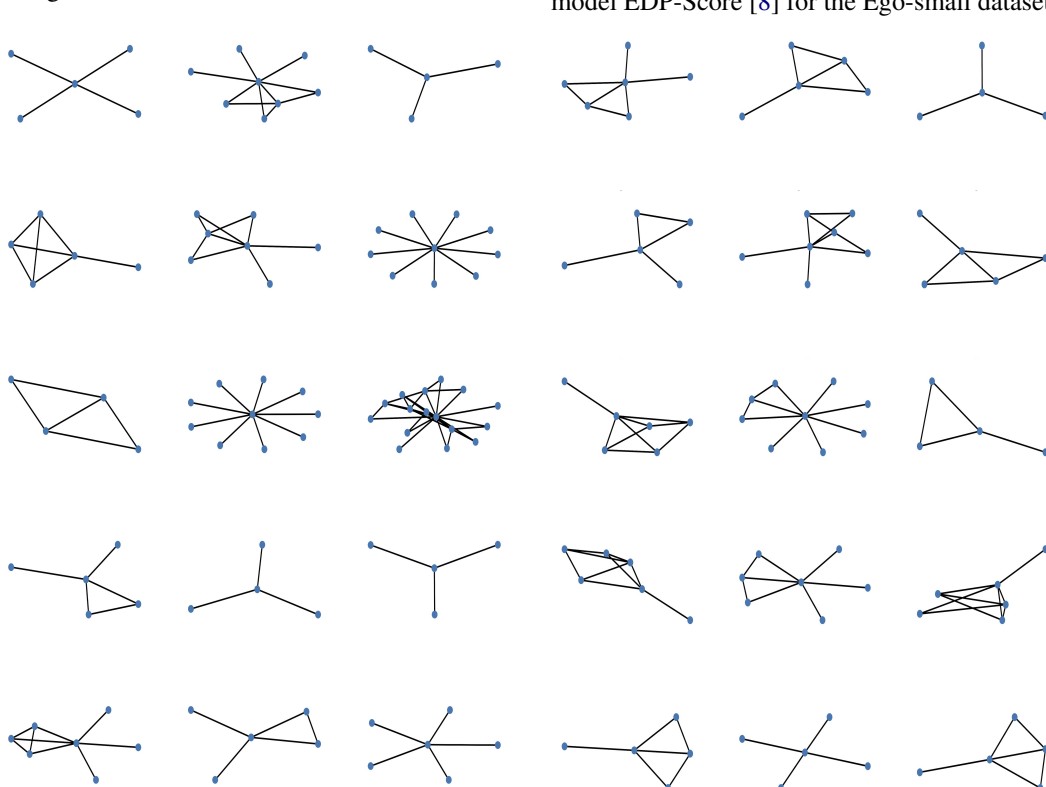

**Figure 4:** Sample graphs generated with the PPGN $L_{\mathrm{vb}}$ model for the Ego-small dataset.

**Figure 5:** Sample graphs generated with the EDP $L_{simple}$ model for the Ego-small dataset.

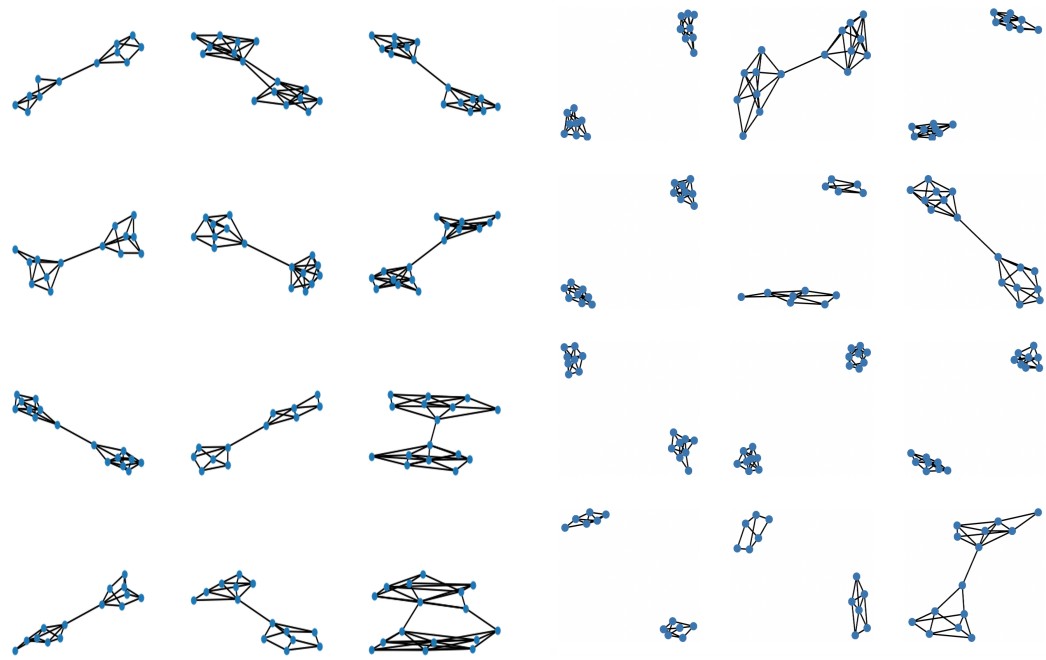

**Figure 6:** Sample graphs from the training set of the Community dataset

**Figure 7:** Sample graphs generated with the model EDP-Score [8] for the Community dataset.

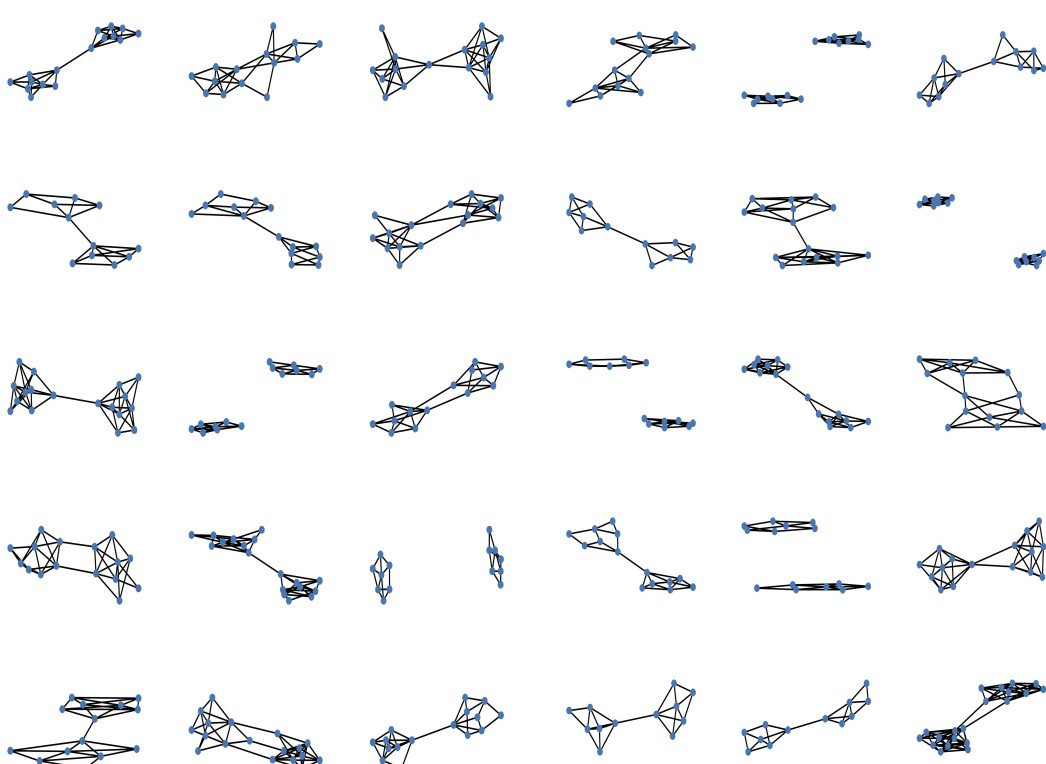

**Figure 8:** Sample graphs generated with the PPGN $L_{\text{vb}}$ model for the Community dataset.

**Figure 9:** Sample graphs generated with the EDP $L_{simple}$ model for the Community dataset.

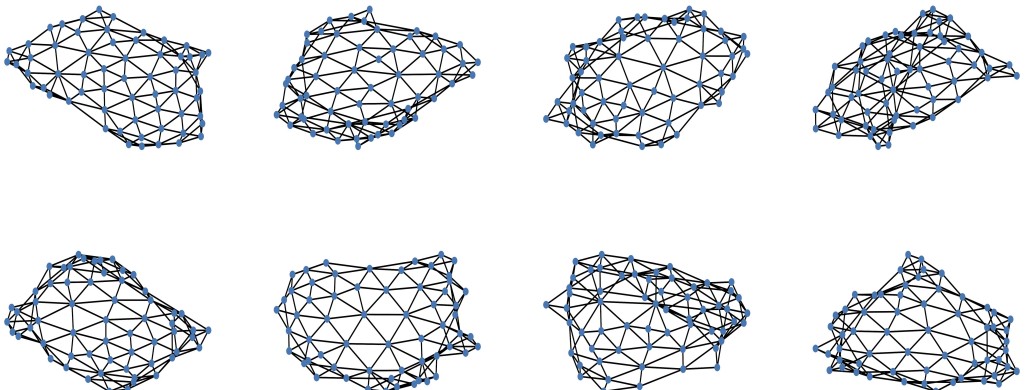

**Figure 10:** Sample graphs from the training set of the Planar-60 dataset.

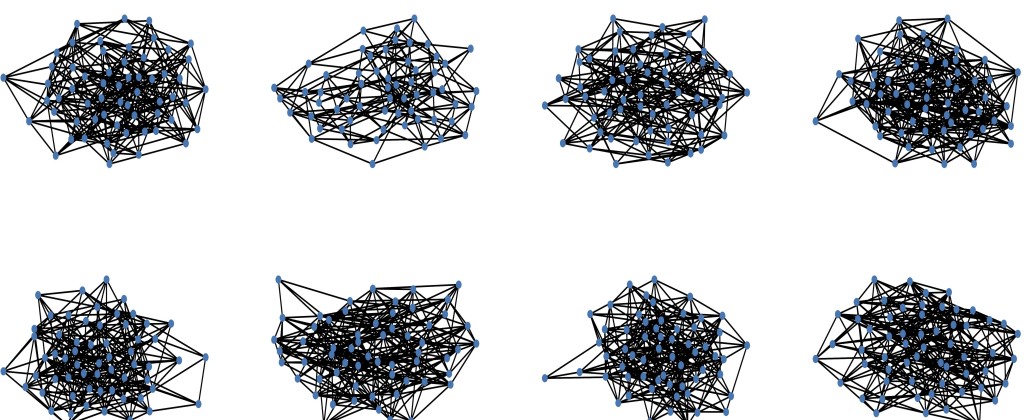

**Figure 11:** Sample graphs generated with the model EDP-Score [8] for the Planar-60 dataset.

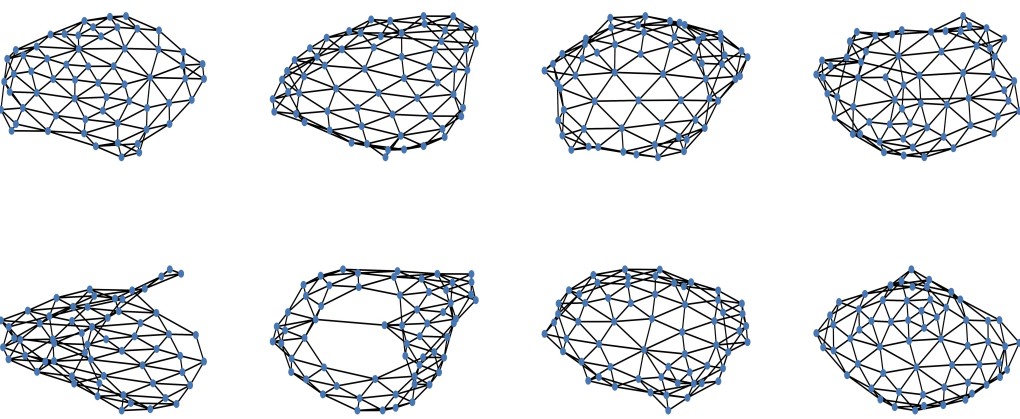

**Figure 12:** Sample graphs generated with the PPGN $L_{simple}$ model for the Planar-60 dataset.

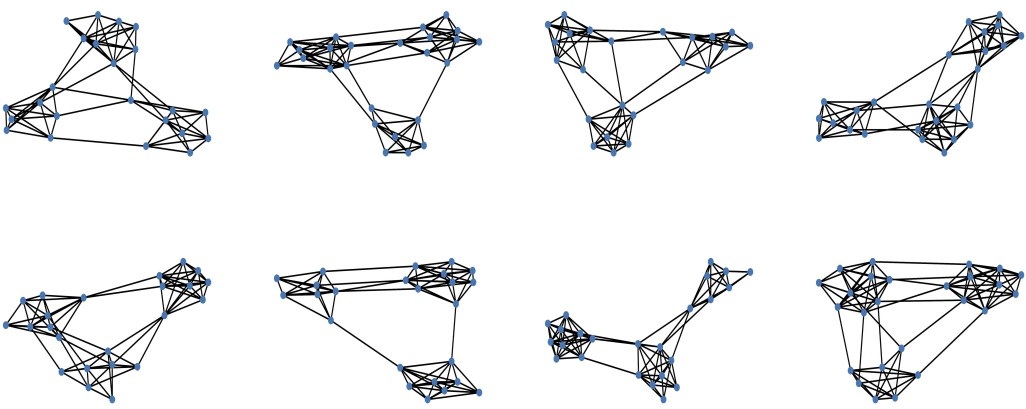

**Figure 13:** Sample graphs from the training set of the SBM-27 dataset.

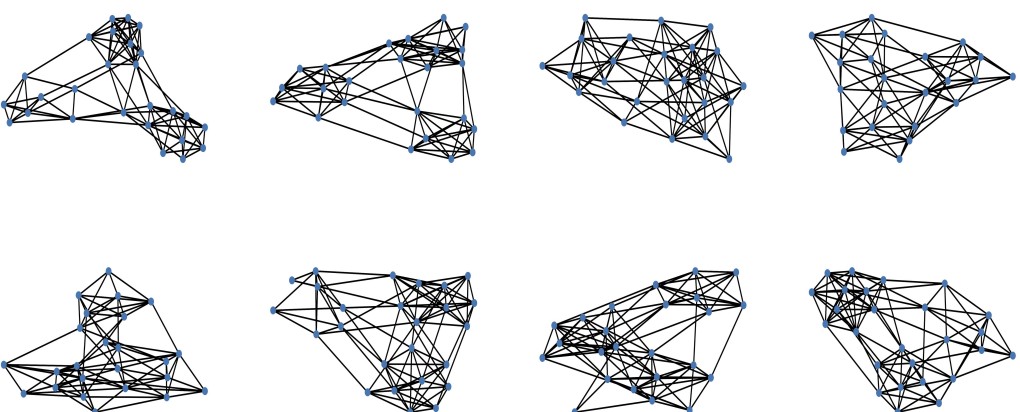

**Figure 14:** Sample graphs generated with the model EDP-Score [8] for the SBM-27 dataset.

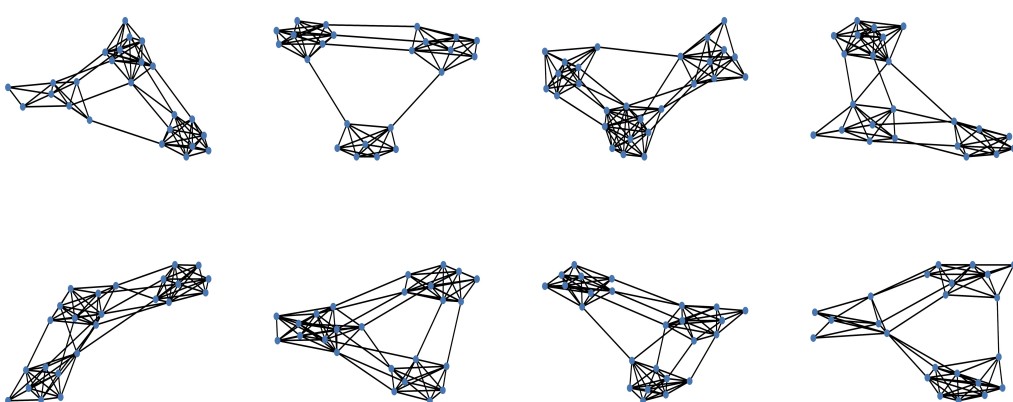

**Figure 15:** Sample graphs generated with the PPGN $L_{simple}$ model for the SBM-27 dataset.

