# OpenReview forum: "Diffusion Models for Graphs Benefit From Discrete State Spaces"
_logconference.io/LOG/2022/Conference — LoG 2022 Poster_

### Official Review · Reviewer_Ff7Y · 2022-10-19

**Overall Score:** 8
**Confidence:** 5

**Review:**

**Summary**

This paper studies the graph generation problem using discrete diffusion models. The paper introduces an approach using discrete noise for the forward Markov process instead of continuous Gaussian noise, guaranteeing that the graph during the process preserves its discrete structure. The paper shows that the discrete noise process can generate graphs with higher quality compared to the previous score-based models.

**Strength**

- The paper is well written with a clear motivation. The related work section provides sufficient background regarding graph generative models and diffusion models.
- The experimental results show that using a discrete noising process instead of a continuous noising process improves the generation performance of EDP-GNN.


**Weakness**

- The proposed discrete process and the loss function are mostly based on [Austin et al. 2021], which is widely used for discrete data such as text.
- The paper claims that using the discrete noising process for EDP-GNN is beneficial, but does not show if it is beneficial for other diffusion processes, such as SDE-score.
- Why using the discrete diffusion is beneficial is not explained in the paper.
- As observed in EDP-score, the generation performance highly depends on the noise schedules of the diffusion process. For a fair comparison between EDP-GNN and the proposed method, the details of the hyperparameter search should be described. If there is some analysis regarding the hyperparameter, the paper would be strengthened.
- The largest dataset is the Planar-60 which are graphs with only 60 nodes. To strengthen the claim, results on a larger dataset are necessary, for example, SBM dataset with 200 nodes as used in [Martinkus et al. 2022] or Grid dataset with up to 361 nodes as used in [Jo et al. 2022].


**Recommendation**

I weakly reject this paper, due to insufficient experiments and analysis. I acknowledge that using discrete diffusion for graph generation is novel, although a concurrent work exists, the proposed discrete process and loss function is mostly based on [Austin et al. 2021] which lacks novelty. If the paper clearly explains why using discrete diffusion is beneficial with extensive analysis, the paper would be strengthened.

**Additional Feedback**

Note that the recent work of [Vignac et al., 2022] also introduces the discrete denoising diffusion for graph generation.

---

** References**

- Austin et al., Structured denoising diffusion models in discrete state-spaces, NeurIPS 2021
- Martinkus, SPECTRE: Spectral Conditioning Helps to Overcome the Expressivity Limits of One-shot Graph Generators, ICML 2022
- Vignac et al., DiGress: Discrete Denoising diffusion for graph generation, Preprint 2022

---

### Official Review · Reviewer_XgPo · 2022-10-21

**Overall Score:** 6
**Confidence:** 2

**Review:**

The paper proposes a generative graph model based on a discrete diffusion process.
The model is evaluated on several benchmarks and the improvements compared to continuous-based formulations are obtained using MMD-based metrics.
It seems that the main contribution of the paper is to take an existing continuous-based diffusion graph generative model, then add the sampling process and a corresponding loss from well-established works on discrete diffusion models (from non-graph domains).
While the results look promising, the paper has some concerns outlined below. Despite some concerns, this work has some interesting insights and given that it's an abstract submission, I recommend acceptance.

### Concerns

- Evaluation metrics for generative graph models have been analyzed and improved recently (see [1,2] below) and the MMD-based graph statistics metrics can be quite noisy and do not always estimate the diversity and fidelity of generated graphs very well.
Moreover, the Deg, Clus and Orb metrics provide different rankings for the models and have different ranges of values, so their averaging (as in Table 1 and 2) can be misleading. For example, based on Deg and Orb metrics, some results of EDP-Score and SDE-Score look better or comparable to the proposed approach. So the methods from [1,2] could provide more accurate estimation.

- It would be useful to see the results for different numbers of denoising steps both for the baselines and proposed model to make more convincing conclusions that the proposed model both improve sampling speed and quality. It could be that the baseline models also perform better with fewer denoising steps. Given that the graphs are quite small, it seems reasonable to have fewer denoising steps.
It would be also useful to evaluate the model on datasets with larger graphs, e.g. as those used in the GRAN paper.

- It's not exactly clear why the discrete formulation would benefit both the quality/diversity of generated graphs and reduce the number of denoising steps. The paper could elaborate on that more.

References:

- [1] On Evaluation Metrics for Graph Generative Models (ICLR 2022)
- [2] Evaluation Metrics for Graph Generative Models: Problems, Pitfalls, and Practical Solutions (ICLR 2022)

### Questions

- To compute the loss in Seq. 3.3 the actual adjacency matrix A is sampled from the probabilistic vectors a_ij, correct? How are the gradients backpropagated through the sampling process?

- Is it possible to just add Gaussian noise to edges (as done in [8]) for the forward process instead of applying Eq. 1 (if I understand correctly, sampling of discrete edges can be done in both cases)? What would be the difference between these two approaches? Is the approach based on Eq. 1 chosen because of some nice theoretical properties or it's also expected to perform better in practice?

Minor:

- L35 can be devised -> devided

---

### Official Review · Reviewer_ySCe · 2022-10-22

**Overall Score:** 6
**Confidence:** 4

**Review:**

The paper introduces a discrete diffusion model for simple graph generation. Unlike previous works, the discrete diffusion model allows the intermediate graphs to be discrete. It has been experimentally shown that using a discrete denoising process can result in higher-quality generated samples, and the denoising process can also be much faster because of fewer denoising steps needed.

Pros:
1. The proposed model is simple and thus easy to follow. The model seems to work well for small-scale graph generation.
2. Two loss objectives are tested.
3. Appendices are comprehensive.

Cons:
1. The forward process is independent for each edge, which ignores the structural information. Besides, a single 2x2 double stochastic matrix is shared by all edges at each step, which is also suboptimal.
2. The proposed model works only for undirected unweighted graphs. In practice, many graphs are weighted or directed. More importantly, the model has been tested only on small graphs (less than 100 nodes), which is unappealing.
3. It is unclear what applications the proposed model can be used for, especially given that it has been only experimented with small graphs.
4. As an extended abstract submission, the related work section is too long. It will be much better to shorten this section and move Appendices A, B, and E to the main text.
5. There are lots of descriptions regarding score matching, I didn't see any connections between it and the proposed model in Sec. 3.

Questions:
1. How can the model be applied to weighted graphs?
2. Will the model work for larger graph generation (e.g., 10K nodes)?

Minor issues:
1. There is a super long sentence in the Abstract (#7-#10), it will be better to rephrase it.

---

### Official Review · Reviewer_9vHG · 2022-10-25

**Overall Score:** 6
**Confidence:** 5

**Review:**

**Contributions of this work**

This work aims to extend denoising diffusion probabilistic models to graph areas. Since the original DDPm model works in continuous space for graph generation  and graph data are in discrete space, this work fille this gap. This work adapts the denoising procedure to an actual graph distribution and uses discrete noise for the forward Markov process in DDPM, which ensures that in every intermediate step the generated graphs in each step are in discrete space.  The major contributions of this work are as follows:
1.This paper proposes to adopt DDPm to generate graphs in discrete space, which fills the gap between the discrete space of a graph and continuous space of DDPM.
2.The experimental results show that using a discrete noising process results in higher quality generated samples with fewer denoising steps, facilitating faster graph generation procedure.

**Strong points of the paper**

1. The graph generation problem is important and hard due to the discrete characteristic of graphs. And DDPM is a promising approach to graph generation.
2. The proposed method is novel since it makes the intermediate graphs in DDPM to be discrete. This is a super proper way for graph generation.
3.The provide experimental results verify the effectiveness of the proposed method.
4. This paper is well organized and easy to follow.

**Questions and Suggestions**

1. The optimization of $L_{vb}(\mathbf{A}_0)$ should be more clear because the optimization in discrete space (or sampling) is hard and it is also one of the hardest challenges of this paper. In the current version, the optimization of the loss function in Page 3 is not clear.
2.  I would suggest adding more experiments on real world graphs.
3. Please add some comments to Algorithm 1 and 2 to illustrate the proposed method more.

**Overall Evaluation**

This topic, graph generation, is an important and changeling problem.This paper proposes an elegenet method to modify DDPM for graph generation in discrete space. The proposed method is novel and technically sound. Thus, I would like to recommend acceptance at this time.

---

### Meta-Review · Area_Chair_KDwP · 2022-11-18

**Confidence:** 4
**Recommendation:** Accept

**Meta Review:**

This paper proposes to use discrete diffusion for graph generation, which is an interesting and novel idea. In addition, the reviewers in general agree that experiments and analyses are sufficient. The response from the authors additionally effectively addressed the remaining concerns of the reviewers which reached a consensus of an acceptance.

---

### Decision · Program_Chairs · 2022-11-22

Accept (Poster)